# Conspecific and Human Sociality in the Domestic Cat: Consideration of Proximate Mechanisms, Human Selection and Implications for Cat Welfare

**DOI:** 10.3390/ani12030298

**Published:** 2022-01-25

**Authors:** Lauren R. Finka

**Affiliations:** 1Battersea Dogs and Cats Home, 4 Battersea Park Rd, Nine Elms, London SW8 4AA, UK; lauren.finka@ntu.ac.uk; 2Brackenhurst Campus, School of Animal, Rural and Environmental Sciences, Nottingham Trent University, Nottinghamshire NG25 0QF, UK

**Keywords:** sociability, wellbeing, stress, group living, domestication, *felis*

## Abstract

**Simple Summary:**

The domestic cat is the only species within the *felis* genus to have transitioned from a wild, solitary species to one of the most popular human-companion animals globally. In stark contrast to their closest wild ancestors, the domestic cat displays an impressive capacity to cohabit successfully with both humans and other cats. However, at an individual level, domestic cats demonstrate substantial variability in their sociability towards both species. Such variability may be influenced by a range of factors including their early life experiences, genetic selection, and individual cat and human characteristics, in addition to various factors associated with their social and physical environment. The impact of these factors may have important implications regarding a cat’s social relationships, their adaptability to various social contexts, and, ultimately, their wellbeing. In line with modern pet-keeping practices, domestic cats may often be exposed to lifestyles which present a range of complex social and environmental challenges, although it is unclear how much cats have been selected by humans for traits that support adaptability to such lifestyles. This review aims to summarise what is currently known about the various factors that may influence domestic cats’ sociality and sociability towards both humans and cats, with a predominant focus on populations managed by humans in confined environments. Current limitations, knowledge gaps, and implications for cat wellbeing are also discussed.

**Abstract:**

Sociality can be broadly defined as the ability and tendency of individuals to reside in social groups with either conspecifics and/or other species. More specifically, sociability relates to the ability and tendency of individuals to display affiliative behaviours in such contexts. The domestic cat is one of the most globally popular companion animals and occupies a diverse range of lifestyles. Despite an arguably short period of domestication from an asocial progenitor, the domestic cat demonstrates an impressive capacity for both intra- and interspecific sociality and sociability. At the same time, however, large populations of domestic cats maintain various degrees of behavioural and reproductive autonomy and are capable of occupying solitary lifestyles away from humans and/or conspecifics. Within social groups, individuals can also vary in their tendency to engage in both affiliative and agonistic interactions, and this interindividual variation is present within free-living populations as well as those managed in confined environments by humans. Considerable scientific enquiry has focused on cats’ social behaviour towards humans (and conspecifics to a much lesser extent) in this latter context. Ontogeny and human selection, in addition to a range of proximate factors including social and environmental parameters and individual cat and human characteristics, have been highlighted as important moderators of cats’ sociability. Such factors may have important consequences regarding individuals’ adaptability to the diverse range of lifestyles that they may occupy. Where limitations to individuals’ social capacities do not enable sufficient adaption, compromises to their wellbeing may occur. This is most pertinent for cats managed by humans, given that the physical and social parameters of the cats’ environment are primarily dictated by people, but that positive human-selection for traits that enhance cats’ adaptability to such lifestyles appears to be limited. However, limitations in the availability and quality of evidence and equivocal findings may impede the current understanding of the role of certain factors in relation to cat sociability and associations with cat wellbeing, although such literature gaps also present important opportunities for further study. This review aims to summarise what is currently known about the various factors that may influence domestic cats’ sociality and sociability towards both humans and conspecifics, with a predominant focus on cats managed by humans in confined environments. Current limitations, knowledge gaps, and implications for cat wellbeing are also discussed.

## 1. Introduction

In a relatively brief period of evolutionary time, the domestic cat has transitioned from a wild solitary species to one of the most popular companion animals globally. During their initial domestication (from wild populations of *F. silvestris lybica* [1]), natural selection pressures are likely to have favoured bolder individuals, as well those with a greater tolerance to human and conspecific proximity [2]. Subsequently, as the value of cats as a source of human companionship increased, a degree of active selection by humans for cat tractability likely followed [3]. However, even within modern-day domestic cat populations, it is unclear of the strength to which (both human and conspecific) sociality has been selected (either naturally or artificially), given that (i) domestic cats may still be motivated to seek out a primarily solitary existence, and can survive under such conditions [4,5,6,7], that (ii) socialised cats from companion populations may live and/or readily interbreed with unsocialised cats from free-living populations, and additionally that (iii) the most intensive period of humans’ cat selection has occurred within the last century, with aesthetic features largely prioritised over traits that might enhance sociability towards conspecifics and/or humans [1,8,9].

Despite these possible constrains to cats’ sociality, the domestic cat is still capable of residing within social groups and may actively choose to associate with conspecifics and/or humans, in each case potentially developing positive social relationships. In both free-living populations, as well as those managed by humans in confined environments, cats may display a range of affiliative behaviours. Affiliative behaviours directed towards conspecifics include vertical tail raising on approach (e.g., the ‘tail up’ signal), as well as initiating various forms of physical contact including nose touching, play, allo-grooming, allo-rubbing, tail wrapping, and sleeping and resting together or in close proximity [10,11,12]. Vocalisations such as purring and meowing between mother and offspring dyads occur frequently and are thought to serve important communicative and care solicitation functions [13,14]. Similar social behaviours are often directed towards humans during cat–human interactions [15,16,17,18,19] and cats are well documented as having the capacity to develop affiliate social relationships with people [20,21,22,23].

At a species level, the domestic cat occupies a diverse range of lifestyles with varying degrees of association with conspecifics and humans [24]. Broad lifestyle categories include cats that can be described as free-living (i.e., feral, street, or stray cats) and those that are living in confined environments managed by humans (e.g., the domestic home, shelter or rehoming centre, and research facility). In many cases, close associations with conspecifics and/or humans may promote health advantages for cats. For individuals under some form of human management, these typically include the provision of primary or supplemental feeding, veterinary care, and access to warm, sheltered, and safe environments. For cats outside of human management, the benefits of living with conspecifics may include communal raising of offspring and shared access to clumped resources of value such as food and shelter. 

At the same time, cats are considered to be ‘socially flexible’ rather than ‘socially obligate’, meaning they possess the potential to adapt to different forms of social living, but that group living (with conspecifics and/or humans) is not necessary for their survival. Degrees of sociability may be highly variable at the individual level, even amongst those occupying similar lifestyles [22,25]. Within lifestyle categories, the availability and quality (i.e., distribution and abundance) of the cats’ physical resources can vary greatly [26,27,28], as can the characteristics (e.g., age and sex, personality, previous experiences) of the cats and/or humans with which individuals cohabit [29,30]. 

Variability in cats’ social behaviour towards both humans and conspecifics, as well as the impacts of cohabitation on their wellbeing, may be influenced by a range of ontogenetic [31] and genetic factors [32] and their potential interaction [33] in addition to the various social and environmental parameters associated with their specific lifestyle [12,34,35,36,37,38]. Thus, in certain instances, close associations with humans and/or conspecifics may be detrimental rather than beneficial to the cats’ wellbeing, especially where individual social flexibility or adaptability is limited [39,40]. These discussion points form the basis of this review, with the intention that such knowledge can be particularly useful in supporting optimal wellbeing outcomes for domestic cats across the various contexts where they are managed by humans. Inherent difficulties associated with the reliable, practical, and valid measurement of ‘wellbeing’ and variation in how it is defined, operationalized, and its measures subsequently interpreted, are well established [41,42,43]. As such, where evidence is considered relevant to cat wellbeing within this review, a broad definition of this term is applied to include any measures or outcomes that might potentially provide useful information about the mental and/or physical health state of a cat.

## 2. Proximate Factors and Their Links to Conspecific Social Behaviour in Free-Living Populations

In free-living populations, cat density and active group living is primarily determined by resource abundance and distribution, with cat densities increasing, and groups typically forming, where resources are plentiful and localised (e.g., refuse areas, farm buildings, or supplemental feeding stations) [26,29,44]. The nature of group-living and conspecific social interactions may be influenced by a range of factors such as relatedness and familiarity, age, sex and sex ratio, and individual personality, in addition to human intervention (e.g., resource provisions and neutering). Where group living occurs, these may often be matrilineal in structure, being comprised of related females, their offspring, and immigrant adult males [29,44,45]. Agonistic interactions amongst free-living individuals are described as rare, but occurring more frequently towards unfamiliar/unrelated cats (i.e., non-group members), particularly those that are female [44,46]. Amongst group-living adults, individuals are also described as having ‘preferred associates’ with which they spend proportionality more time in proximity to and perform affiliative behaviours towards [11,29,44,47]. Amongst offspring, observations of related juveniles and kittens suggest individuals prefer to associate with conspecifics of a similar age category to themselves and littermates over non-littermates [29]. 

Rates of agonistic behaviour between group members have been explained as a function of variation in age and sex ratios in males, and of neuter status in both males and females. For example, in a (presumably intact) colony with larger male to female and adult-male to juvenile-male ratios, adult males were reported to initiate aggressive behaviour towards other adult males more often than other sex-age classes [29]. In contrast, in a colony where adult male to female and adult-male to juvenile-male classes were smaller, adult males were observed to initiate aggressive behaviour most frequently towards juvenile males [29]. In a study of group-living populations of female cats located around supplemental feeding stations in Israel [48], intact females were found to engage in significantly larger proportions of agonistic interactions with conspecifics (e.g., chasing, hiss, yowl and growl, threat approach, stare, and physical attacks) than neutered females. Similar associations with neuter status and social interactions were reported in populations of group-living cats in Italy [49], where the neutering of both males and females was associated with a significant decrease in rates of aggression (e.g., striking with a paw, biting, assuming threatening postures, chasing, ritualized vocal duels, and physical fighting), but also a reduction in conspecific proximity. Although differences in pre- and post-neutering frequencies of affiliative behaviours were not significantly different overall, rates of nose sniffing and rubbing were noted to decrease in frequency for most dyads, with the exception of two specific male-male dyads, where rates increased. Collectively, these results would suggest that neutering may generally decrease the tendencies for individuals to engage in agonistic interactions (and social interactions in general), although for some specific dyads, affiliative interactions may increase. 

Amongst free-living cats, personality differences may also influence the general behavioural styles of individuals towards conspecifics, although there is limited research within this area. In a study of unneutered male cats across several colonies [50], cats were described as having either predominantly ‘proactive’ or ‘reactive’ personalities. ‘Proactive’ males were defined as those engaging in more agonistic (e.g., threating, fighting, and chasing), but also affiliative (e.g., ‘nose sniffing, passive contact, and rubbing’), interactions with other group members. In contrast, ‘reactive’ individuals were mostly characterised by displaying avoidant (e.g., ‘avoiding, crouching, flying, hissing’) and less agonistic behaviour. 

## 3. Proximate Factors and Their Links to Conspecific Social Behaviour in Confined Populations

For cat populations managed in confined environments, group-living and group composition are primarily dictated by humans, and, thus, cats have little choice over this aspect of their sociality. Despite this, cats may still display individual variation in the conspecifics they choose to associate with or avoid [51,52]. Variations in the nature of conspecific social interactions have been linked to similar factors to those highlighted in free-living populations, although these relationships and their direction of effects are not consistent across studies. For example, in a USA-based survey of cat owners that had recently introduced a cat into their household [53], the provision of outdoor access (but not cat age, sex, or cat group size) was associated with increased rates of fighting amongst cats. In contrast, a more recent (USA) study [12] based on a substantially larger population of cats and owners reported that age (i.e., younger cats), sex (i.e., female), multi-cat group size (i.e., larger), as well as the recent addition of an unfamiliar cat to the home, were all associated with increased rates of cat conflict, whilst the provision of outdoor access was not. Similar trends were reported in a UK survey [54] where again, provision of outdoor access was not predictive of increased aggression, but sex and neuter status (i.e., neutered females) were. However, it is important to note here (and elsewhere in the review where results of owner-completed surveys are mentioned), that owner reports of cat’s behaviour obtained via this method are subjective and potentially contain various sources of bias. Their results should thus be interpreted with this in mind, especially where survey data are compared between studies. 

Little is currently known about the impact of variations in environmental provisions on the social dynamics of multi-cat groups residing in the various different confined environments that are managed by humans. Nonetheless, the relative quantity and distribution of resources are considered central components in the management and prevention of conspecific conflict [55,56,57]. Their importance is certainly plausible, given the impact of variations in resource abundance and distribution on cat sociality in free-living populations [44]. Despite this, in previous research [12], no significant relationships were identified between the frequency of (group level) conspecific conflict and the quantity of various resources provided in the home (e.g., scratching posts, food stations and litterboxes, outdoor access, available indoor space). However, greater quantities of litter boxes and food stations were associated with increased frequencies of affiliative behaviours. While this study provides some useful ‘top level’ insights into the relationships between conspecific social behaviour and resource provision, further detailed studies on this topic are needed, and across different confined environments. These should include the consideration of pertinent variables such as resource quality (i.e., size and suitability) and relative distributions, in addition to individual cat characteristics such as age, sex, neuter status, relatedness, and personality. Investigations should also account for intra-group variation in dyadic social relationships and consider their interactions with these other factors, rather than quantifying social behaviour at the group level [12].

Initial behavioural responses during introductions are considered important determinants of the future relationships between cohabiting cats. For example, the same study [12] reported greater rates of conflict and lower rates of affiliative behaviours between cats in households where owners indicated initial introductions ‘did not go well’, with similar results reported in [53]. In general, gradual methods of cat introductions are considered to help promote more amicable conspecific relationships. Such methods incorporate scent swapping between cats before progression to (slowly increased) periods of visual, and then supervised, physical access. However, while these methods are widely recommended [56,58,59], their benefits are currently based on anecdotal observations rather than empirical research. In the one study where current rates of fighting were assessed relative to methods of cats’ initial introductions (i.e., either ‘gradual’ or ‘immediate’) [53], no significant differences were reported between owners using either method. However, the author highlighted issues associated with the way that ‘gradual’ and ‘immediate’ methods were categorised, potentially resulting in less than meaningful statistical comparisons. For example, ‘gradual’ introductions encompassed a broad range of time periods, from hours to weeks, and did not include any details of the specific methods used (e.g., scent swapping or visual access initially). Further study into the potential benefits of gradual approaches during cat introductions and their impact on long-term relationships are therefore warranted. 

Familiarity, relatedness, and social exposure during the sensitive period (i.e., approximately between 2–7 weeks of age) have also been linked to differences in conspecific social behaviour. In observations of mostly unrelated cat dyads within the domestic home, longer lengths of cohabitation were found to negatively correlate with rates of conspecific aggression [60]. In observations of cat dyads housed together in a cattery [10], litter mates (i.e., related and together since birth) spent more time in physical contact with each other and were more likely to feed together and allo-groom, compared to unrelated pairs (that had previously lived together in a home for at least a year). Rather than simply their genetic relatedness, the authors attributed these greater rates of affiliative behaviours to littermates having been socialised to each other during their sensitive periods and then also experiencing an extended duration of cohabitation due to their remaining together into adulthood. In a large group of confined cats residing in a private property, duration of cohabitation was similarly positively associated with proximity and allo-grooming between conspecifics [61]. In this latter study, however, where duration of cohabitation was controlled, related individuals (but not necessarily littermates) performed affiliative behaviours more frequently than unrelated individuals, suggesting that both duration of cohabitation and general relatedness may be important modifiers of conspecific social behaviour. To date, it is unclear which of these factors (i.e., genetic relatedness, cohabitation during the sensitive period, total duration of cohabitation), and their interactions, may have most impact on the social behaviour of cats towards conspecifics as adults. However, it is anticipated that genetically related cats that reside together during their sensitive period and have subsequently lived together for longer periods (within a stable environment) might be more likely to have long-term affiliative social relationships with each other compared to other classes.

Within the domestic home, rates of agonistic behaviour have been linked with aspects of cat’s personality. In [54] a (weak) positive relationship between owner ratings of their cat’s level of fearfulness and conspecific aggression (e.g., growls, hisses, bites, scratches) was reported. In [12] a (weak) positive relationship between cats described as ‘sedentary and shy’ and rates of conflict behaviours (e.g., ‘flee’, ‘hiss’ and ‘twitch tail’) was also identified.

## 4. The Impact of Cohabiting with Conspecifics in Confined Environments on Cat Wellbeing

Given the potential links between increased group size and conspecific conflict [12], it is logical to assume that living in larger multi-cat groups may be associated with more negative wellbeing outcomes for cats. However, systematic reviews on this topic highlight a lack of cross-study consensus within both the domestic home [38] and shelter environments [36]. In the home, for example, greater numbers of cohabiting cats have been significantly linked to greater rates of owner-reported ‘behaviour problems’ and anxiety [62] and increased house soiling [28,63,64,65], but also fewer ‘behaviour problems’ [66], lower Cat Stress Scores or CSS (a posture and behaviour based scoring system) [67,68], and less negative interactions with humans [69,70,71]. Additionally, several studies have reported no significant links between cat group size and ‘behaviour problems’ [72], house soiling [69,71], obesity [71], or physiological stress [30,73].

These equivocal findings potentially arise due to several pertinent factors. Issues associated with the variability of study methodologies and limitations in their designs and analytical approaches mean that the relative effects of cat group size may not be clear and/or comparable between studies, making it difficult to reach a consensus [38]. Furthermore, the varied, multi-faceted nature of housing and husbandry conditions to which domestic cats are exposed, and subsequently studied under, in addition to the potential diversity of the individual characteristics of the cats within each multi-cat group, means that cat populations may be very demographically diverse. This may occur both across separate study populations, but also within a single comparison group, within a single study [38]. This diversity makes it difficult to isolate the impact of an individual variable, such as group size, on cat wellbeing, without considering and (where suitable) controlling for the influence of important cat- and environment-based covariates. For example, a study focused on cats in the domestic home [74] highlights the importance of individual cat characteristics and their potential moderating role in the relationship between group size and increased adrenocortical activity (thus physiological stress), measured via faecal glucocorticoid metabolites (GCM). While in this study, no main effects of cat group size on variation in GCM were detected, several age-related, within-group effects were identified. For cats housed in groups of 3–4, those aged 2 years or older had significantly higher GCM values than cats under 2 years of age. Younger cats from single cat households also had higher GCM concentrations than those housed in groups of 3–4. These findings suggest that the relative impact of cat group size on physiological stress may be age dependent, with younger cats having lower, and older cats higher, adrenocortical activity when housed in groups. However, relationships between GCM and stressors can vary depending on the nature of the stressor, [75,76], sampling method, and time period [77], making it difficult to interpret GCM values in relation animal wellbeing, particularly when these are considered in isolation to other relevant physiological and behavioral measures [77,78]. 

In the cattery environment, similar limitations regarding evidence quality and a lack of cross-study consensus were evident [36]. Findings from this review also highlighted the importance of both cat and environment features when investigating links between group size and cat wellbeing. For example, in one study, no significant differences were identified where Cat Stress Scores (CSS) were compared between cats housed singly or with one or two familiar conspecifics, although the CSS of these cats were significantly lower compared to cats housed in large, unfamiliar groups [79]. In another study, singly housed cats in barren conditions were found to have significantly higher CSS compared to cats housed in more enriched environments, either alone or in a large group of unfamiliar conspecifics [80]. In a third study [39] comparing CSS between cats housed singly and those housed in a group with unfamiliar cats, no differences were found for cats considered socialised towards conspecifics, although CSS were higher for cats considered unsocialised. Additionally, when ‘unsocialised’ cats were added to group housing, their presence caused the CSS of other group members to increase. Relative environmental provisions, the socialisation status of the individual, and that of cohabiting conspecifics, may therefore determine whether residing with conspecifics is more, less, or similarly stressful for the individual than being housed alone within a cattery context.

## 5. Proximate Factors and Their Links to Cat–Human Social Interactions in Confined Environments

Several studies have highlighted various human characteristics as important determinants of cats’ social behaviour during human-cat interactions (HCI). Observational studies taking place in the domestic home [81,82] suggested cats demonstrate preferences for social interactions with adults (particularly females) over children. These differences in cats’ responses may be explained by variations in humans’ interaction styles, given that children (in particular males) may be more likely to approach resting cats, pick them up, and follow retreating cats than adults, behaviours which are likely to be perceived as threatening by the cat, or to at least induce a degree of discomfort [81,82]. In contrast, adults (in particular women) may be more likely to vocalise to cats and crouch down to their level, postures and behaviors which may be perceived as a less threatening and more encouraging of the cat to engage in social interactions [82]. In [34] observations of HCI with adult owners in the home suggested that cats interacted with owners for longer durations when the interactions were initiated by the cat as opposed to the owner. Additionally, cats were found to be more likely to engage in interactions when initiated by the owner, if owners were generally responsive to the cats’ requests for interactions [34]. 

Regions of the cats’ body that are touched by humans may also impact the nature of their behavioural responses. Human stimulation to the cats’ caudal region may produce greater rates of human-directed aggression (e.g., hissing, biting, smacking, or scratching) as well as behaviours indicative of discomfort (e.g., flattening the ears, flicking and/or swishing the tail) [83,84]. In contrast, stimulation to cat’s temporal regions may lead to greater rates of affiliative behaviours (e.g., closing or half-closing the eyes, “kneading” with paws, purring, rubbing against the human, and dribbling), with stimulation to the perioral regions, flank, stomach, and back producing much greater between-individual variation in responses [83]. Understanding of the impact of human behaviour styles on cat comfort and social behaviour during HCI (incorporating findings from the aforementioned studies) was recently formalised into a set of ‘best practice’ guidelines for humans [85]. Compared to control HCI, cats were found to exhibit significantly more affiliative and positively valanced behaviour (e.g., tail waving, kneading, sniffing and rubbing, ears forwards), as well as less agonistic behaviours (e.g., hiss/growl, cuff/swipe, bite) and fewer signs of conflict (e.g., tail swishing, ears rotated/flattened, paw lift, rapid groom, head/body shake, freeze/crouch, avoid/move/turn away), when humans followed the guidelines [85]. 

In addition to general styles of interaction, certain facial and postural cues displayed by humans may promote more positive social responses from cats. For example, companion cats were found to approach unfamiliar humans significantly more often when they performed a ‘slow-blink sequence’ towards the cat, rather than when they adopted a neutral expression [86]. Cats were also found to spend a longer time in contact with their owners when they displayed a ‘happy’ rather than an ‘angry’ posture and facial expression [87].

In a study examining the underlying structure of human-cat interactions taking place in owners’ homes [17], links with owner personality were identified. HCI were reported to be less patterned and structured where owners scored higher for the personality trait Neuroticism. In contrast, owners scoring higher in Conscientiousness had more complex styles of interactions with their cats, meaning the structure of their interactions involved a more diverse range of cat and owner behavioural elements. Owner mood has also been linked to differences in intent to interact with the cat [88,89]. For example, owners rating themselves as more anxious and touchy were found to display greater intents to initiate interaction, whilst those that were more extroverted or depressed showed less intents [89]. In the same study [89], the more extroverted and agitated the person was, the more the cat was found to approach the person during an ongoing HCI, whereas the more numb the owner reported they were, the less the cat approached. In another study by the same authors [88], cats were found to head and flak rub more during interactions with owners that reported a more depressed mood. 

The personality of owners has also been linked to more general, longer-term aspects of cats’ human-directed social behaviour [37], with the direction of results sharing parallels with those identified within the parent–child [90,91] and owner-dog dynamics [92,93]. In a large UK-based survey of owners and their cats [37], higher owner Neuroticism was associated with more aggressive and fearful cat behavioural styles as well as greater reported ‘problem behaviours’. In contrast, higher owner Conscientiousness, Agreeableness, and Openness were associated with less aggressive and aloof cat behavioural styles, and higher Conscientiousness with less fearful, but more gregarious, styles. 

However, across these various studies [17,37,88,89], specific details of the handling styles exhibited by owners during HCI and/or their associations with the cats’ behavioural responses were either not quantified or reported, making the results hard to interpret in relation to the cats’ experience and comfort during HCI. Further studies investigating the relationship between human personality/mood, their HCI styles and cats’ subsequent reactions are therefore warranted.

In addition to owner personality, various characteristics of the cat and their environment may impact on the nature of their social interactions with humans. For example, in a survey of Brazilian cat owners [35], several risk factors were associated with increased rates of human-directed aggression across various contexts (i.e., when petted, when startled, during play, and when around unfamiliar people and animals). These factors included the cat living in a household described by their owners as ‘frenetic’ (i.e., busy and unpredictable), the cat being of mixed breed rather than a pedigree, being described as “disliking” rather “liking” being petted, and having “poor” relationships with other animals in the household. 

Cats’ individual characteristics (i.e., see further sections) may also affect their perceived value of human-social interaction relative to other non-social stimuli categories such as food and toys. For example, preference tests performed on cats within a shelter center context indicated substantial variation in their preferences, and while overall, more cats preferred interacting with humans, a substantial sub-population were found to prefer the non-social stimuli [94].

## 6. The Impact of Cohabiting with Humans in Confined Environments on Cat Wellbeing

During typical social interactions with humans, a cats’ exposure to a single HCI that it finds aversive may induce an acute (and thus potentially short lived) negative experience (see previous section). In contrast, the repeated, frequent exposure to aversive HCIs is much more likely to induce chronic (and thus longer term) negative states within individuals, potentially leading to their compromised wellbeing. While this line of scientific enquiry remains largely uninvestigated, a preliminary study [74] in the domestic home reported that cats that were described by their owners as generally “tolerating” being stroked had higher faecal glucocorticoid metabolite levels, compared to cats that were described as actively “liking” or “disliking” being petted. Such findings highlight the potential impact of a cats’ experiences during petting on their physiological stress response, with cats “tolerating” petting potentially at greater risk of increased stress. However, further research, which considers a range of additional (suitable) behavioural and physiological measures, is warranted in order to understand the implications of aversive HCI to cats’ wellbeing more broadly. 

In the previous section, busier, less predictable households [35] and owners scoring higher in Neuroticism [37] were associated with greater owner reports of cat human-directed aggression [35,37] and ‘problem behaviors’ and anxiety [37]. These human-based factors have also been associated with several other cat health- and welfare-linked outcomes, supporting their potential validity as risk factors for cat wellbeing. 

For example, in [73], cat urinary cortisol concentrations were positively correlated with the total number of humans living in a home, as well as human density. In another study [30], cats from single households were found to have higher faecal glucocorticoid metabolites where their owners reported being more socially active with other humans. Such findings might suggest that greater human presence and levels of human-social activity within the home create more stressful environments for cats, leading to their compromised wellbeing. However, again, more detailed investigations are required to explicitly test this hypothesis.

In [37], cats of more neurotic owners were reported to display greater stress-linked ‘sickness behaviours’ (a composite score representing poorer coat condition, greater frequencies of cystitis, vomiting, diarrhoea, and constipation). These cats were also more likely to have an existing medical condition, to be of an unhealthy weight, and were more likely to have either restricted or no access to the outdoors. In humans, Neuroticism has been associated with decreased empathy, more authoritarian and over-protective parenting, as well as the provision of harshly controlled, but poorly structured, environments [95,96,97]. It is therefore important to determine if similar dynamics might be present in human-cat relationships, given that harsher, more forceful, and less predictable formal handling and husbandry styles have been associated with more negative wellbeing outcomes for cats [98,99]. While comparable studies in the domestic dog [100,101] reported similar owner-Neuroticism–pet-wellbeing relationships to those in [37], the current evidence base remains primarily correlational rather than causal (and based on subjective owner reports). Therefore, further investigations examining the causal mechanisms underpinning the relationships between owner personality, human-cat interactions and cat wellbeing are needed.

## 7. The Ontogeny of Human and Conspecific Sociability in Cats Managed by Humans in Confined Environments

Positive human-social experiences during early developmental periods appear important predictors of friendliness (and its generalisation) towards humans later in life. Two studies [102,103] have investigated the effect of kittens being handled by humans during different periods of their development (e.g., from 1–5 weeks of age, or from 2–6 weeks, 3–7 weeks, or 4–8 weeks), and for different amounts of time each day (e.g., 15 min versus 40). The authors reported that kittens handled within the 2–7 week time period were generally more amenable to being handled and were quicker to approach people, as were those that were handled for longer periods each day. This 2–7 week ‘sensitive period’ was identified as the stage where kittens were deemed most receptive to social learning regarding humans, with handling commencing towards the later stages of this period reported to produce less effective results. Increased handling following a specific ‘socialisation and habituation’ programme (as opposed to more basic, limited handling) within a shelter context was also found to produce cats that were reported as displaying less fear-based behaviours towards their adoptive owners, as well as providing them with more ‘social support’ when assessed at a year of age [31]. In addition to the timing, quality, and quantity of handling, the number of different handlers may also impact on kittens’ subsequent behaviour towards unfamiliar humans. In one study, kittens regularly handled by five different people (as opposed to a single person or not handled at all) were observed to make fewer attempts to retreat from a stranger [104]. 

Potentially heritable traits may also interact with kittens’ early social experiences to impact the nature of their social behaviour towards humans. In the following studies [102,103], kittens considered to be more ‘timid’ were anecdotally noted as being initially less amenable to handling, although this did improve if they were then adequately socialised to humans during 2–7 weeks of age. A study of cats residing in a research facility [33] suggested the genetic influence of paternal ‘boldness’ might enhance the degree to which (human socialised) offspring demonstrate sociability, via a greater receptibility to human-socialisation. In [33] while kittens that had been regularly handled between 2 and 12 weeks of age and sired by unfriendly, less ‘bold’ fathers were friendlier than unsocialised kittens from both friendly and unfriendly fathers, those that were socialised and sired by friendly males were the most friendly and confident of all groups. When tested at a year of age, these latter cats were most likely to approach and explore novel objects, approach and interact sociably with a person, and were also least likely to behave aggressively when approached and handled. Such results would suggest these genetic/ontogenetic effects also demonstrate temporal stability. 

Interactions between heritable maternal traits, early social exposure to humans, and subsequent human-sociability have not been investigated, although it is likely that similar relationships are present. However, genetic (inherited) maternal influences are more difficult to study practically, given the challenges of controlling for maternally induced epigenetic effects that may impact kittens during their pre- and post-natal periods, prior to weaning. For example, modifications to kittens’ gene expressions which affect their behavioural phenotypes may occur as a result of maternal stressors, maternal deprivation, and the dynamics of the mother-offspring relationship [105].

Cats’ social experiences with conspecifics during their sensitive period appear similarly important in influencing subsequent social behaviour towards other cats. In a study of kittens within a research facility [106], it was noted that kittens’ behavioural responses to an unfamiliar kitten varied depending on whether they had been reared with/without their mother and with/without their siblings until six weeks of age. Kittens were regularly tested from 2 to 20 weeks of age and kittens that were reared in isolation from their littermates (either with or without their mother) were found to display more frequent biting, wrestling, approaching, lowering their ears, as well as sideways stances with piloerect fur, when in an unfamiliar kitten’s presence. These littermate-deprived kittens were also described by the authors as less able to engage in appropriate social play with a conspecific, and that the nature of their social interactions was more agonistic than play-like. These kittens were also reported to have their claws extended more often and were less able to display bite inhibition. For example, such kittens were described as intensifying their biting and attacking when the test kitten performed a distress cry or tried to escape, compared to the kittens reared with their littermates, who would usually retreat at this point. The kittens reared without littermates were also described as generally being more ‘hyperactive’ when exposed to both objects and the test kittens. 

Other studies suggest that a cats’ early familial environment may impact their future behaviour towards conspecifics as well as humans. In another lab-based study [107], groups of kittens were either separated from their mother and litter mates at 2, 6, or 12 weeks of age. Based on experimenter observations of the cats over a 9-month period, the kittens separated from their mothers at 12 weeks of age were anecdotally described as being more ‘docile and friendly’ towards conspecifics as well as humans, while the kittens isolated at 2 weeks of age were described as generally more anxious and likely to behave aggressively when frightened. Anecdotal observations from other studies suggest that the presence of both the mother [108] and littermates [109] during socialisation towards humans may reduce anxiety and increase the kittens’ receptivity to humans’ attempts to socialise them. Additionally, in a large survey of pet cats and their owners [110], cats that were (retrospectively) reported as being removed from their mothers earlier (i.e., before 8 weeks of age) were more likely (as adults) to behave aggressively towards unfamiliar humans than those weaned later (i.e., between 12–15 weeks of age). The authors also reported that cats that either had not been separated from their mothers at all, or had not been separated until reaching adulthood (i.e., at least one year or older), were less likely to behave aggressively towards both conspecifics as well as familiar and unfamiliar people. 

Under certain conditions, such as limited amounts of human-socialisation and barren environmental conditions, impacts of a cats’ early social environment on their subsequent behaviour towards humans may be subsumed by paternal genetic effects. In a study of cats within a research facility [111], differences in kittens’ latency to approach a person, durations they held their tail above the horizontal, and level of tolerance to human restraint during blood sampling were assessed. When tested at 20 weeks of age, no significant differences were reported in kittens’ behaviour relative to whether they had experienced early separation from their mother (i.e., 5 weeks) and subsequent individual caging with either basic (i.e., 15 mins three times a week for 3 weeks) or no human handling, or whether they had experienced later separation from their mother (i.e., at 6–7 weeks) and been subsequently housed with conspecifics. However, variation in kitten’s behaviour during tests was associated with the identity of their father. Kittens sired by two particular fathers were reported as behaving more sociably towards humans and being more tolerant to restraint, whilst kittens sired by other fathers were reported to be less sociable and more aggressive. While it is certainly plausible that these behavioural differences amongst kittens could be due to differences in the boldness/shyness amongst their fathers [33,112], no data on paternal personality were reported in the study. 

In general, adequate social exposure towards humans during the cats’ sensitive period (i.e., 2–7 weeks of age) and beyond, appear central to the development and stability of human-sociability, although inherited traits (e.g., boldness/shyness) may also play important mediating roles. Later separation (i.e., beyond the cats ‘sensitive period’ for human-socialisation) from mother and siblings may also enhance both conspecific and human-cat social relationships later in life. Related conspecifics that remain together from birth may be more likely to engage in affiliative interactions than non-related cohabiting conspecifics. However, whether cats have a similar ‘sensitive period’ for the development of sociability to conspecifics, if this generalises to unfamiliar, unrelated cats in the same way that it might in humans, and the role of heritable personality traits as moderating factors remain unclear. 

## 8. Lifestyle Variation, Social Flexibility, and Welfare Considerations

Within a cats’ lifetime, some individuals may transition from an independent free-living lifestyle to cohabiting with humans (and also conspecifics) within the confines of a domestic home. In some instances, the cat might actively facilitate this transition, for example they may start spending time near to human dwellings and eventually decide to ‘move in’. In most cases, however, this process occurs due to human intervention. Typically, a cat deemed to be ‘stray’ or ‘unowned’ is physically removed from its original location, temporarily housed in a shelter environment, before then being placed into a domestic home of humans’ choosing. Indeed, such cats can represent a large proportion of the total shelter population, with a UK study suggesting up to as many as 42% within a given year [113]. It is worth noting that prior to a period of free living, some of these stray or unowned cats may have originated from domestic homes, and thus been appropriately socialised (to humans and potentially conspecifics) during their sensitive period. However, many of these free-living cats may have had no, or very limited, social experiences with either species, and thus potentially a much more limited capacity to adapt well to future environments that require close cohabitation with them. 

Within the shelter environment, a cats’ previous social experiences (and inferred degree of socialisation) appear to be important determinants of their wellbeing. In [39] cats that were deemed to be ‘unsocialised’ towards humans had higher Cat Stress Scores (CSS) than those considered to be ‘socialised’, and those considered ‘unsocialised’ to conspecifics had higher CSS when housed in groups. In another study [40], cats that had previously resided in single, rather than multi-cat, homes were found to have significantly higher CSS during the first few days of their arrival at a shelter, even though they were housed individually. A lack of previous socialisation to humans and conspecifics is therefore likely to compromise the wellbeing of cats when housed within a shelter environment. While the impact of conspecifics on cats unsocialised to cats can potentially be mitigated by proving them with single housing, an equivalent is obviously more difficult to provide for cats unsocialised to humans within such human-managed confined environments. Comparable data regarding the wellbeing of cats within the domestic home, based on their lack of previous socialisation to humans and/or cats, is missing. However, it is anticipated that similar relationships are likely present, with such previously unsocialised cats struggling more (or generally failing) to adapt.

Attempts to human-socialise unfriendly and fearful cats that are already outside of their ‘sensitive period’ (i.e., 2–7 weeks of age) can be common practice within shelters [114,115]. While the aim of this process may be to try to enhance a cats’ capacity to adapt well to future cohabitation with humans, such practices may pose important ethical, as well as practical, considerations. The cats’ early developmental or ‘sensitive period’ is characterised by enhanced neurobiological plasticity, outside of which neuronal circuits are typically less susceptible to modification via experience [116]. Additionally, innate responses to human proximity in non-socialised cats are primarily characterised by fear, attempted avoidance, and defensive behaviour [117]. Therefore, both the facilitating of non-fear inducing and positive social experiences with humans, and their ability to be rapidly generalised to other humans and contexts, is likely to be more difficult to achieve, beyond this initial period of plasticity. Instead, it is likely that acceptance of humans in previously non-socialised, fearful cats occurs through the process of stimulus-flooding. This process involves the exposure of an individual to negative, acutely stressful stimuli from which it cannot avoid, until cessation of the initial behavioural response (i.e., fearful or aggressive reactions) is achieved [118]. Rather than producing a positive end result, this process can potentially induce the experience of ‘learned helplessness’ within individuals, due to their lack of perceived ability to control their exposure to the fear-inducing stimulus [118,119,120]. It is uncertain if cats that have undergone such stimulus-flooding processes are then able to develop generalised positive associations with humans in the same way as cats socialised towards humans during their ‘sensitive’ period are. However, anecdotal reports suggest these cats tend to struggle to cope with domestic living and also typically fail to meet owner expectations for companionship (Oral communication, Battersea Dogs and Cats Home). This may subsequently lead to reduced owner satisfaction [121,122] and potentially cat relinquishment [123,124]. 

However, given the potential moderating role of individual personality (i.e., inherited traits) during cats’ initial socialisation to humans [33,103,111], it is possible that individuals of a certain temperament may be more receptive to developing positive relationships with people, even when these commence outside of their ‘sensitive period’. For example, very bold, free-living, food-motivated cats that are not innately fearful of humans (sensu [3], see next section) may have a greater capacity for positive human-social learning. However, their subsequent adaptability to domestic living may still be limited, considering their lack of previous experience with this lifestyle and its associated challenges [28,125].

## 9. Domestication, Selection, and Implications for Sociability in Modern Day Domestic Cats

In a comparative review of sociality across Felidae, [126] the author notes that while increases in sociality within a species are usually associated with increases in brain to body mass ratios (i.e., encephalisation), this relationship has not been evident during the domestication of the cat. At a genomic level, however, notable differences between domestic cats and their wild progenitors are evident. In a comparative analysis of the domestic cat, European (*F. silvestris silvestris*) and Eastern (*F. silvestris lybica*) wildcat genes [3], positive selection for enriched neural-crest-related genes within domestic cat samples was identified. In mice, these genes are linked with a reduction in fear responses, enhanced memory, and the ability to learn based on positive rewards such as food [3,127,128,129]. These genetic signatures potentially explain the cognitive mechanisms through which initial domestication and increased sociality were able to occur, with less fearful individuals more readily able to tolerate human proximity and create positive associations with humans. 

Compared with their early domesticates, however, modern-day companion cats are likely to experience more socially complex and potentially challenging cat-caregiver dynamics [23,37,66,130]. Despite this, it is unclear whether much active selection for traits that enhance successful adaption to modern human-social relationships and domestic living have been undertaken [2]. For example, in domestic dogs, genetic signatures suggestive of intense selection for prosocial traits such as those associated with enhanced responsiveness towards humans, attention-seeking, and initiation of prolonged social contact are evident [131,132,133]. While in domestic cats, this area of research has received less attention, similar evidence of genotypic-phenotypic relationships for prosocial behavior and their positive human-selection are currently lacking. 

For example, in [32], genetic surveys on microsatellite polymorphisms in a population of domestic cats were conducted, focusing on those linked to receptors (associated with oxytocin release) and previously linked with sociability in other species [131,134]. While in [32], positive associations between microsatellite length and the caretaker-rated friendliness of cats were identified, pedigree cats were found to have *shorter* alleles compared to non-pedigree cats, implying a less enhanced genetic predisposition for human-sociability in pedigree populations. Additionally, while levels of oxytocin and cortisol have previously been found to increase in domestic dogs following social interactions with their owners [135], a recent preliminary study in domestic cats reported that both oxytocin and cortisol levels were actually decreased following human-social interactions [136]. 

In a recent survey of cat owners [25] investigating the heritability of cat behaviour traits, considerable diversity in the social tendencies of cats across different breeds was reported, with no obvious trends towards increased human or conspecific sociability, compared to non-pedigrees. Certain breeds also scored particularly highly for unsociable behaviours, including limited contact with people (e.g., British short hair) and aggression towards both conspecifics and humans (e.g., Turkish van). Therefore, rather than enhancing human and/or conspecific sociability, in certain cases, selective breeding practices may have produced the opposite effect. 

It is, however, important to highlight that the data used to assess cats’ behaviour in these studies were based on the subjective reports from owners and caretakers and from two distinct populations of cats and owners (i.e., Finnish [25] and Japanese [32]). Breed types were also confirmed by owners [25] or vets [32] rather than via objective genetic or morphometric methods. Given that humans’ perceptions of different cat breeds and their behaviour seem to vary between studies and demographic populations [137,138,139], and that only a limited number of pedigree cats (*n* = 40) and breed types (*n* = 10) were included in the genetic analysis conducted in [32], the generalisability of these findings may be limited. Further studies that incorporate standardised behavioural observations, in addition to more comprehensive genetic analyses, are likely to facilitate a more comprehensive understanding of the potential differences in sociability between pedigree and non-pedigree cats at both the behavioural and genetic level.

Little is known about whether sociability towards humans and conspecifics might share similar trait pathways within domestic cats. However, latent behavioural variables derived from various surveys of owners/caretakers reporting on cats suggest that human-directed and cat-directed social behaviours form their own separate structures, and, as such, could be considered individual traits [140,141,142]. Additionally, while in [25], some breeds demonstrated similar rates of expression of particular social behaviours towards both species, this was not consistent across all breeds. For example, the Turkish Van had a comparatively high probability of aggression towards both cats and humans, whilst the Devon Rex had a comparatively high probability of aggression towards humans, but comparatively low probability towards other cats. 

The apparent emphasis on the physical characteristics of pedigree breeds (over traits that may convey social advantages) might have other limiting consequences regarding their social capabilities. For example, it is possible that breed-based characteristics that cause discomfort or limit the physiological or communicative functionality of an individual may also negatively impact their social relationships with both humans and/or conspecifics. For example, highly brachycephalic breeds such as the modern Persian and Exotic short hair can experience a range of health conditions associated with their eyes, skin, and respiration [143,144]. In the Scottish Fold, their characteristic ‘folded’ ears are the result of selection for heritable gene mutations, which also cause abnormal bone and cartilage development, leading to chronic pain and mobility issues [145]. The presence of such chronic conditions has the potential to induce poor mood, increased irritability, and human- and conspecific-directed aggression, in addition to a range of other behaviours which owners tend to find problematic [146,147,148]. Such individuals may also be more likely to occupy sedentary lifestyles [144] and generally have a reduced desire or tolerance for positive social interactions with humans and/or conspecifics. 

Breeds with highly exaggerated morphology may also struggle to effectively communicate during social interactions. A recent study [149] has suggested that breeds with more extreme facial features, such as those that are very brachycephalic (e.g., Exotic short hairs, modern Persians) or dolichocephalic (e.g., Oriental short-hair and Sphynx), may have more limited abilities to produce clearly identifiable and differentiable facial expressions. Given the importance of facial expressions during the expression of emotions and intentions [150,151], and thus the maintenance of social relationships, such limitations might have important consequences regarding the nature of cats’ social interactions. Links between cats’ altered morphology and their communicative abilities may also extend beyond the face. For example, the shortened limbs and proportionally elongated spine of the Munchkin may limit their ability to effectively alter their posture for communicative purposes, while the largely absent tail of the Manx cat prohibits their ability to perform important affiliative behaviours such as conspecific tail wrapping and the vertically raised ‘tail up’ signal [11,48,152]. Further studies are required to fully understand how breed-linked morphology might impact on the behavioural repertoires utilised by individuals during social interactions with both humans and conspecifics, and the subsequent effect this might have on the nature of their social relationships. Likewise, research is needed to assess the potential negative impact of breed-linked health conditions on cats’ tendencies to engage in different styles of social interaction (e.g., affiliative, agonistic, tolerant, or avoidant) with both species. 

## 10. Conclusions

At a species level, the domestic cat displays an impressive capacity to cohabit successfully with both humans and conspecifics, despite their recent asocial ancestry and apparent limited selection for traits that might enhance their social capabilities. At the individual level, however, cats may demonstrate substantial diversity in their human and conspecific sociability. This may have an impact on their ability to cope with the various social challenges to which they are exposed when under human management. This diversity may largely be explained by interactions between early social experiences and inherited behaviour traits, although further studies which more broadly and explicitly test these hypotheses are needed. Additionally, various proximate factors, including individual cat and human characteristics, appear important mediators of cats’ social interactions with both species. However, the availability and quality of evidence varies, and in some instances equivocal findings limit current understanding.

As urbanisation continues and companion animals are increasingly relied upon for social and emotional support [23,153], domestic cats are likely to be exposed to increasing environmental and social challenges whilst cohabiting with humans and conspecifics in confined environments [125]. Thus, active selection for traits that enable cats to adapt to these lifestyles are likely to be beneficial for their wellbeing, as well as their future relationships with both species. Additionally, adequate exposure of cats to positive social experiences during their early development, in combination with suitable housing and handling practices, represent important aspects of their husbandry that may support similar outcomes.

## Data Availability

Data sharing not applicable.

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
