# Peer review of "Conspecific and Human Sociality in the Domestic Cat: Consideration of Proximate Mechanisms, Human Selection and Implications for Cat Welfare"

_animals, 2022, doi:10.3390/ani12030298_

Round 1

Reviewer 1 Report

Thank you for inviting me to review this review manuscript about cats’ sociality (with conspecific or human) in relation to the possible mechanisms involved as well as implications on their wellbeing. Overall, I enjoy reading it – the review is well written, mostly well-organised and does a great job in summarising the state of art about cat sociality in different contexts.

In general, I understand that the author is summarising the works that have been done in understand cat's sociality (toward conspecific or humans) at different contexts. The author did a great job on this. However, when we talk about proximate causes, highlighting the factors alongside some possible explanations about the 'how' a factor may led to a cat being social or not would be helpful. For example, in section 5, I am not understanding very well how the behaviours of humans (adult vs. children, owner or not) affected HCI…how does vocalising makes a cat interact more if a kid picks up a cat?So, there are a few places where I found not entirely clear (to me) and hope that the author could clarify.

Line 120-122. is this information about the agonistic interactions between groups? The information here states that such behaviours are ‘rare’ and toward unfamiliar individuals (esp. females), but the next para talks about within-group agonistic behaviours, mostly from male to male.

Line 132 – okay…. the transition between the two factors are a bit muddle…even though the topic sentence has stated this para would be able the age/sex ratio within group as well as neuter status of an individual… I wonder whether revising the topic sentence to something more general or reorganising information will make the rest of the section easier to be read. Just some thoughts about this section: 1) place information about personality first (because it is also about between-group interactions and it will mirror the structure of the next section (3) about individual variation);  2) a general statement about within-group agonistic behaviours in free-ranging cat populations vary. Followed by a sentence about factor 1 (ration of age/sex) with examples, and another statement about factor 2 (neuter status) with examples?

Revise reference in text – from Gajdoš Kmecová et al. (2019) to Kmecová et al. (2019)

Line 216, it will be useful for reader to understand sensitive period is ‘approximate 2-7 weeks’ (line 225) is mentioned here first.

Line 271 what is GCM measures? Stress level? And higher GCM means…? Please add a few words to clarify

Line 336-342 – the information in this para seem to relate to the first para (i.e., more general HUI style), perhaps it shall be organised in the same para?

Question to the author regard section 5 and 6: how does the author define ‘well-being’ in general? does the behaviours mentioned in section 5 with cats that have problematic behaviours or ‘not liking’ to be petted reflect that they are not happy? And this ‘not very happy’ actually relates to their well being too? If so, how does section 5 and 6 differ in terms of the section content?

Author Response

R1:

Thank you for inviting me to review this review manuscript about cats’ sociality (with conspecific or human) in relation to the possible mechanisms involved as well as implications on their wellbeing. Overall, I enjoy reading it – the review is well written, mostly well-organised and does a great job in summarising the state of art about cat sociality in different contexts.

This is much appreciated – thank you!

In general, I understand that the author is summarising the works that have been done in understand cat's sociality (toward conspecific or humans) at different contexts. The author did a great job on this. However, when we talk about proximate causes, highlighting the factors alongside some possible explanations about the 'how' a factor may led to a cat being social or not would be helpful. For example, in section 5, I am not understanding very well how the behaviours of humans (adult vs. children, owner or not) affected HCI…how does vocalising makes a cat interact more if a kid picks up a cat?So, there are a few places where I found not entirely clear (to me) and hope that the author could clarify.

Many thanks for highlighting this – I have edited the text in several places [ln 361-366] to provide more clarity and context in relation to the likely causal mechanisms

Line 120-122. is this information about the agonistic interactions between groups? The information here states that such behaviours are ‘rare’ and toward unfamiliar individuals (esp. females), but the next para talks about within-group agonistic behaviours, mostly from male to male.

Line 132 – okay…. the transition between the two factors are a bit muddle…even though the topic sentence has stated this para would be able the age/sex ratio within group as well as neuter status of an individual… I wonder whether revising the topic sentence to something more general or reorganising information will make the rest of the section easier to be read. Just some thoughts about this section: 1) place information about personality first (because it is also about between-group interactions and it will mirror the structure of the next section (3) about individual variation);  2) a general statement about within-group agonistic behaviours in free-ranging cat populations vary. Followed by a sentence about factor 1 (ration of age/sex) with examples, and another statement about factor 2 (neuter status) with examples?

I agree these paragraphs and their structure were a bit confusing -  thank you for highlighting this – I have edited the whole of section (2), lines 139-192, so that hopefully it is clearer now. As all the studies mentioned deal with intra-group interactions (this is now more clearly stated), and personality is a comparatively standalone factor to the others mentioned, I have chosen to keep this as a final and separate point.

Revise reference in text – from Gajdoš Kmecová et al. (2019) to Kmecová et al. (2019)

The in-text refs have been corrected to Vancouver style

Line 216, it will be useful for reader to understand sensitive period is ‘approximate 2-7 weeks’ (line 225) is mentioned here first.

Added (ln 256)

Line 271 what is GCM measures? Stress level? And higher GCM means…? Please add a few words to clarify

For clarity, this text has been edited to:

Ln 317- 321 “For example,  a study focused on cats in the domestic home (71) highlights the importance of individual cat characteristics and their potential moderating role in the relationship between group size and increased adrenocortical activity (thus physiological stress), measured via faecal glucocorticoid metabolites (GCM).”

Line 336-342 – the information in this para seem to relate to the first para (i.e., more general HUI style), perhaps it shall be organised in the same para?

Thanks for this suggestion, I’ve more explicitly stated that the Haywood et al study leads on from the previous ones mentioned in this section, which hopefully clarifies why it is included at the end of this section:

Ln 381: “Understanding of the impact of human behaviour styles on cat comfort and social behaviour during HCI (incorporating findings from the aforementioned studies) were recently formalised into a set of ‘best practice’ guidelines for humans, designed to provide cats with greater autonomy and increased comfort (82)”.   

Question to the author regard section 5 and 6: how does the author define ‘well-being’ in general? does the behaviours mentioned in section 5 with cats that have problematic behaviours or ‘not liking’ to be petted reflect that they are not happy? And this ‘not very happy’ actually relates to their well being too? If so, how does section 5 and 6 differ in terms of the section content?

Thanks for pointing this out – a very broad definition of wellbeing has been included in the abstract, (ln 61-65) although this does not help to resolve the current comments, where I agree that there is theoretically overlap between some of the evidence reported in section 5, and that included in section 6. I have thus added in an additional section at the beginning of 6 to acknowledge this:

Ln 446-464: “During typical social interactions with humans, a cats’ exposure to a single HCI that it finds aversive may induce an acute (and thus potentially short lived) negative experi-ence (see previous section). In contrast, the repeated, frequent exposure to aversive HCIs is much more likely to induce chronic (and thus longer term) negative states within individ-uals, potentially leading to their compromised wellbeing. While this line of scientific en-quiry remains largely uninvestigated, a preliminary study (71) in the domestic home re-ported that cats that were described by their owners as generally “tolerating” being stroked had higher faecal cortisol metabolite levels (and thus greater adrenocorticoid ac-tivity) compared to cats that were described as actively “liking” or “disliking” being pet-ted. Such findings highlight the potential impact of a cats’ experiences during petting on their physiological stress response, with cats “tolerating” petting potentially at greater risk of increased stress. However, further research which considers a range of additional (suit-able) behavioural and physiological measures is warranted, in order to understand the implications of aversive HCI to cats’ wellbeing more broadly.  

In the previous section, busier, less predictable households (35) and owners scoring higher in Neuroticism (37) were associated with greater owner reports of cat human-directed aggression (35, 37) and ‘problem behaviours’ and anxiety (37). These human-based factors have also been associated with several other cat health and welfare-linked outcomes, supporting their potential validity as risk factors for cat wellbeing.”

Reviewer 2 Report

This is very well written review clearly showing gaps in  the field of human-cat interactions. I have few suggestions which can be considered by the author prior to acceptance.

the most intensive period of humans’ cat selection has occurred within the last century, with aesthetic features largely prioritised over traits that might enhance sociability towards conspecifics and/or humans (e.g. see Morris  1999, Driscoll et al 2007 and 2009, Plitman et al 2019) – perhaps pleiotropic effect can be mentioned here

citations in the text should be improved according to instructions for authors

Personality differences may also influence the general behavioural styles of individ- 149 uals towards conspecifics. In a study by Natoli et al (2005)- pozriem ci nie je tych prac viac

Rather than simply their genetic relatedness, the authors attributed these greater rates of affiliative behaviours to littermates having been socialised to each other during their sensitive periods (i.e. approximately 2-7 weeks of age) and then also experiencing an extended duration of cohabitation due to their remaining together into adulthood.- is there any experimental study attempting to discover whether is this sensitive period more important than genetic itself relaredness? If not, this issue can be also highlighted. I think that Curtis et al 2003 did not manipulate all these variables (i.e. relatedness x sensitive period).

L. 266-281 I believe that personality traits can also be associated with perceived stress in cat and coul contribute to mixed results of reviewed studies.

What about cat size? It was found that reproductive success of male feral cats in urban environment positively correlates with their body weight (Natoli et al. 2007), thus I assume that interactions with other cats can be influenced also by differences in body mass.

Natoli, E., Schmid, M., Say, L., & Pontier, D. (2007). Male reproductive success in a social group of urban feral cats (Felis catus L.). Ethology, 113(3), 283-289.

L. 304-310 – this suggests that experiences play crucial role in cat preferences for women. Is there any experimental study examining cat with no prior experiences in interactions with humans?

L. 318-334. Please mention also the role of hormones in social interactions. It seems that social interaction decreases oxytocin in cats (but opposite was found in dogs) and cortisol. Please see here:

Nagasawa, T., Ohta, M., & Uchiyama, H. (2021). The urinary hormonal state of cats associated with social interaction with humans. Frontiers in Veterinary Science, 8, 680843.

L. 336-354 please consider also:

Johnson, E. A., Portillo, A., Bennett, N. E., & Gray, P. B. (2021). Exploring women’s oxytocin responses to interactions with their pet cats. PeerJ, 9, e12393.

Reiger and Turner 1999, Turner and Rieger- please check whether Reiger or Rieger is correct

L. 544: Cat Stress Scores (CSS)) – please check number of parentheses

Author Response

R2:

This is very well written review clearly showing gaps in  the field of human-cat interactions. I have few suggestions which can be considered by the author prior to acceptance.

Many thanks and for the useful comments - hopefully these have all been sufficiently addressed

the most intensive period of humans’ cat selection has occurred within the last century, with aesthetic features largely prioritised over traits that might enhance sociability towards conspecifics and/or humans (e.g. see Morris  1999, Driscoll et al 2007 and 2009, Plitman et al 2019) – perhaps pleiotropic effect can be mentioned here

This is a really good point and I totally agree that it’s a potentially relevant one to make. However,  I’m cautious about expanding on the current size of the review any further, given its already substantial length, and as all the other points I touch upon here in th4 into are expanded upon in detail in later sections, for balance, I feel I would need to do the same regarding pleiotropy and mostly in a speculative way given the limited evidence. For these reasons I would prefer not to add this point in.

citations in the text should be improved according to instructions for authors

These have now been corrected to Vancouver throughout

Personality differences may also influence the general behavioural styles of individ- 149 uals towards conspecifics. In a study by Natoli et al (2005)- pozriem ci nie je tych prac viac

I have added  (ln 185) “although again there is limited research within this area” to this sentence

Rather than simply their genetic relatedness, the authors attributed these greater rates of affiliative behaviours to littermates having been socialised to each other during their sensitive periods (i.e. approximately 2-7 weeks of age) and then also experiencing an extended duration of cohabitation due to their remaining together into adulthood.- is there any experimental study attempting to discover whether is this sensitive period more important than genetic itself relaredness? If not, this issue can be also highlighted. I think that Curtis et al 2003 did not manipulate all these variables (i.e. relatedness x sensitive period).

Great point - I’ve edited the text accordingly:

Ln 274-279: “To date, it is unclear which of the factors including genetic relatedness, cohabitation dur-ing the sensitive period, total duration of cohabitation and their interactions, may have most impact on the social behaviour of cats towards conspecifics as adults. However, it is anticipated that genetically related cats that reside together during their sensitive period and have subsequently lived together for longer periods (within a stable environment) might have be more likely to have long-term affiliative social relationships with each oth-er, compared to other classes.”    

  1. 266-281 I believe that personality traits can also be associated with perceived stress in cat and coul contribute to mixed results of reviewed studies.

Thanks great point, and this is actually relevant to individual cat characteristics more generally -  I’ve edited the text accordingly so that it now reads as:

Ln 309: “Additionally, the varied, multi-faceted nature of housing and husbandry conditions to which domestic cats are exposed and subsequently studied under, including the potential diversity of the individual characteristics of the cats within each multi-cat group means that cat populations may be very demographically diverse. This may occur both across separate study populations, but also within a single comparison group, within a single study (Finka and Foreman-Worsley 202138). This diversity makes it difficult to isolate the impact of an individual variable such as group size, on cat wellbeing, without considering and (where suitable) controlling for the influence of important cat and environment-based covariates.”

What about cat size? It was found that reproductive success of male feral cats in urban environment positively correlates with their body weight (Natoli et al. 2007), thus I assume that interactions with other cats can be influenced also by differences in body mass.

Natoli, E., Schmid, M., Say, L., & Pontier, D. (2007). Male reproductive success in a social group of urban feral cats (Felis catus L.). Ethology113(3), 283-289.

Thank you for highlighting this. As the review focuses on the factors that specifically influence social living and to a greater extent the sociability of cats (i.e the ability and tendency of individuals to display affiliative behaviours in such contexts), I would argue that the link between body mass and reproductive success is slightly outside of its remit. Although in the above study, these factors are also reported to relate to the ‘dominance rank’ of individuals (a value derived from the outcome of observations of agonistic/submissive interactions), I would argue that such a latent construct is theoretically and operationally different from the ‘sociability’ of an individual (although these two constructs might be related to a degree), thus for simplicity I would prefer not to include mention of this paper in order to maintain the logic and consistency of the review. If the paper had reported links between size/body weight and rates of agonistic and/or affiliative behaviours directed towards conspecifics, I would definitely agree that the results would be relevant to mention. As far as I am aware, there are no studies that have investigated specifically how body mass has influenced either group living or conspecific sociability (as per the above definition). 

  1. 304-310 – this suggests that experiences play crucial role in cat preferences for women. Is there any experimental study examining cat with no prior experiences in interactions with humans?

Not that I’m aware of. The difficulty here would be that to test this, such cats would have to have had no prior socialisation to humans, which if this was the case, would likely result in them being highly fearful and avoidant of all humans, regardless of human sex/gender. Ethically, this sort of study would also be questionable.

  1. 318-334. Please mention also the role of hormones in social interactions. It seems that social interaction decreases oxytocin in cats (but opposite was found in dogs) and cortisol. Please see here:

Nagasawa, T., Ohta, M., & Uchiyama, H. (2021). The urinary hormonal state of cats associated with social interaction with humans. Frontiers in Veterinary Science8, 680843.

Ah, thank you for highlighting this study -  I have included it where I felt it was most relevant:

Ln 709-719: “For example, Arahori et al (2017)in (32) conducted genetic surveys on microsatellite polymorphisms in a population of domestic cats were conducted, focusing on those linked to receptors (associated with oxytocin release) and previously associated linked with sociability in other species ( Kis et al 2014128, Romero et al 2014,131 ). While Arahori et al (2017) (32) found positive associations between microsatellite length and the caretak-er-rated friendliness of cats, pedigree cats were found to have shorter alleles compared to non-pedigree cats, implying a less enhanced genetic predisposition for human-sociability in pedigree populations. Additionally, while levels of oxytocin and cortisol have previous-ly been found to increase in domestic dogs following social interactions with their owners (132), a recent preliminary study in domestic cats reported that both oxytocin and cortisol levels were actually decreased following human-social interactions (134).”

  1. 336-354 please consider also:

Johnson, E. A., Portillo, A., Bennett, N. E., & Gray, P. B. (2021). Exploring women’s oxytocin responses to interactions with their pet cats. PeerJ9, e12393.

Thank you for highlighting this very interesting paper – as the focus of the review in on the factors influencing the cats’ social behaviour and experience during HCI (rather than the humans) I feel that this paper it is slightly outside of its remit.

Reiger and Turner 1999, Turner and Rieger- please check whether Reiger or Rieger is correct

Corrected

  1. 544: Cat Stress Scores (CSS)) – please check number of parentheses

corrected

Reviewer 3 Report

The manuscript is clear, well-written, and is an important contribution to the field of cat social behavior. I enjoyed reading the paper.

The manuscript could use improvement in a few locations to further clarify the text. Please find my comments below.

Major Comments

  1. The information could be structured in a way that is more accessible. For example, text could be condensed and a table outlining the major findings of the review could be included.

  2. In several locations (e.g., L165, L208 & L248) you mention the results of owner surveys. Owner surveys can be very valuable, however it should be considered that the results of these studies are based on the owner’s perception of the cat’s behavior, and not a direct measurement of the cat’s behavior. 

Minor Comments

  1. I don't know the exact requirements for the journal, but the abstract seems long and it is missing a simple summary. 

  2. L53-54: Socialized cats can also be found living in free-ranging populations. You mention this in L534 but I think it can be mentioned here briefly. 

  3. At L72 consider including Behnke et al. here (who measured human-directed allorubbing) and at L73 add Edwards et al. who also studied cat-human attachment. 
    1. Behnke et. al (2021). The effect of owner presence and scent on stress resilience in cats. Applied Animal Behaviour Science https://doi.org/10.1016/j.applanim.2021.105444 
    2. Edwards et al. (2007). Experimental evaluation of attachment behaviors in owned cats. Journal of Veterinary Behavior. https://doi.org/10.1016/j.jveb.2007.06.004 

  4. L115-116: Although colonies can be matrilineal in structure (related females, offspring, and immigrant males), this is not the only structure of domestic cat social groups. Groups can also be comprised of primarily adult males (as you point out in L128) with some of these groups consisting of mostly neutered adult males. I think it is important to point out that group living in domestic cats can take many forms. Free-roaming cat groups may vary by cat sex ratio, genetic relatedness, sexual status, and level of human care.

  5. L336: Somewhere in this section you may also want to mention cats  prefer different forms of human interaction (e.g., playing and petting are highly preferred while solely being talked to is less preferred: Vitale Shreve et al., 2017, Social interaction, food, scent or toys? A formal assessment of domestic pet and shelter cat preferences)

  6. L600: Please clarify the statement "...constraints associated with domestic cats’ sociality at a species level.” What "constraints" are you referring to? I question this because if we compare cat sociality to domestic dog sociality, how do they really differ? Both species display highly individual social behavior (in dogs, Barrera et al.,2010, Responses of shelter and pet dogs to an unknown human) and social behavior in both species is flexible and dependent on experiences during the sensitive period. The only constraints I could see are those around length of sensitive period (which is longer in the domestic dog than domestic cat) but it is unclear to me how brain/body mass relate. Please elaborate on this a little more.

  7. L619-621: With the statement, “...similar evidence of genotypic-phenotypic relationships and their positive human-selection is currently lacking in domestic cats” I would prefer the wording be changed. To my knowledge, no work has been conducted in this area. So it is less about lacking evidence and more about lacking research.

Author Response

R3:

The manuscript is clear, well-written, and is an important contribution to the field of cat social behavior. I enjoyed reading the paper.

Thank you – much appreciated!

The manuscript could use improvement in a few locations to further clarify the text. Please find my comments below.

Major Comments

  1. The information could be structured in a way that is more accessible. For example, text could be condensed and a table outlining the major findings of the review could be included.

I definitely agree that the review is quite a long one and incorporates a lot of information from many evidence sources as well as some detailed discussion around them, making some of its points potentially less accessible. I’ve had a good think about how including a table might work, but I feel it would likely need to be a rather large one to be useful, which I’m concerned would only ultimately make the paper even larger/denser. I also think that the discussion points around the studies mentioned are quite important for context, detail which is mostly lost where such data are entered into a tabular format. For these reasons I would prefer not to add in a table, although I do appreciate the point.

  1. In several locations (e.g., L165, L208 & L248) you mention the results of owner surveys. Owner surveys can be very valuable, however it should be considered that the results of these studies are based on the owner’s perception of the cat’s behavior, and not a direct measurement of the cat’s behavior. 

This is a really important point – thank you for highlighting – this has now been addressed:

Ln 211: “However, it is important to note here (and elsewhere in the review where results of owner completed surveys are mentioned) that owner reports of cat’s behaviour obtained via this method are subjective and potentially contain various sources of bias. Their results should thus be interpreted with this in mind, especially where survey data are compared between studies.”

Minor Comments

  1. I don't know the exact requirements for the journal, but the abstract seems long and it is missing a simple summary. 

Thank you for highlighting – a simple summary has now been added:

Simple summary:

Ln 13: “The domestic cat is the only species within the felis genus to have transitioned from a wild, soli-tary species to one of the most popular human-companion animals globally. In stark contrast to their closest wild ancestors, the domestic cat displays an impressive capacity to cohabit success-fully with both humans and other cats. However, at an individual level, domestic cats demon-strate substantial variability in their sociability towards both species. Such variability may be influenced by a range of factors including their early life experiences, genetic selection, individ-ual cat and human characteristics, in addition to various factors associated with their social and physical environment. The impact of these factors may have important implications regarding a cat’s social relationships, their adaptability to various social contexts, and ultimately their well-being. This is most pertinent to cats managed by humans in confined environments such as the domestic home, rescue/rehoming centre or research facility, given that these contexts are pri-marily dictated by humans.  In line with modern pet keeping practices, domestic cats may often be exposed to lifestyles which present a range of complex social and environmental challenges, although it is unclear how much cats have been selected by humans for traits that support their adaptability to such lifestyles. This review aims to summarise what is currently known about the various factors that may influence domestic cats’ sociality and sociability towards both hu-mans and cats, with a predominant focus on populations managed by humans in confined envi-ronments. Current limitations, knowledge gaps and implications for cat wellbeing are also dis-cussed”.

  1. L53-54: Socialized cats can also be found living in free-ranging populations. You mention this in L534 but I think it can be mentioned here briefly. 

This has been added and so now reads as:

Ln 81: “(ii) socialised cats from companion populations may live and/or readily interbreed with unsocialised cats from free living populations,”

  1. At L72 consider including Behnke et al. here (who measured human-directed allorubbing) and at L73 add Edwards et al. who also studied cat-human attachment. 
    1. Behnke et. al (2021). The effect of owner presence and scent on stress resilience in cats. Applied Animal Behaviour Science https://doi.org/10.1016/j.applanim.2021.105444 
    2. Edwards et al. (2007). Experimental evaluation of attachment behaviors in owned cats. Journal of Veterinary Behavior. https://doi.org/10.1016/j.jveb.2007.06.004 

Thank you for highlighting  – these have now been included Behnke et. al (2021) to ln 100 and Edwards et al. (2007) to ln 101

  1. L115-116: Although colonies can be matrilineal in structure (related females, offspring, and immigrant males), this is not the only structure of domestic cat social groups. Groups can also be comprised of primarily adult males (as you point out in L128) with some of these groups consisting of mostly neutered adult males. I think it is important to point out that group living in domestic cats can take many forms. Free-roaming cat groups may vary by cat sex ratio, genetic relatedness, sexual status, and level of human care.

This whole section has been reworded for clarity of the various points being made:

Ln 143-159: “The nature of group-living and conspecific social interactions may be influenced by  a range of factors such as relatedness and familiarity, age, sex and sex ratio and individual personality, in addition to human intervention (e.g. resource provisions and neutering). Where group living occurs, these may often be matrilineal in structure, comprising of re-lated females, their offspring and immigrant adult males (42, 41, 29). Agonistic interac-tions amongst free-living individuals are described as rare, but occurring more frequently towards unfamiliar/unrelated cats (i.e. non-group members), particularly those that are female (43, 41). Amongst group-living adults, individuals are also described as having ‘preferred associates’ with which they spend proportionality more time in proximity to and perform affiliative behaviours towards (41, 29, 44, 11). Amongst offspring, observa-tions of related juveniles and kittens suggest individuals prefer to associate with individ-uals of a similar age category to themselves and littermates over non-litter mates (29).”

  1. L336: Somewhere in this section you may also want to mention cats  prefer different forms of human interaction (e.g., playing and petting are highly preferred while solely being talked to is less preferred: Vitale Shreve et al., 2017, Social interaction, food, scent or toys? A formal assessment of domestic pet and shelter cat preferences)

Thanks great point – mention of this paper has been added to end of this section:

Ln 438: “Cats’ individual characteristics (i.e. see next section) may also affect their perceived value of human-social interaction relative to other non-social stimuli categories such as food and toys. For example, preference tests performed on cats within a rehoming center context indicated substantial variation in their preferences, and while overall more cats preferred interacting with humans, a substantial sub-population were found to prefer the non-social stimuli (91).” 

  1. L600: Please clarify the statement "...constraints associated with domestic cats’ sociality at a species level.” What "constraints" are you referring to? I question this because if we compare cat sociality to domestic dog sociality, how do they really differ? Both species display highly individual social behavior (in dogs, Barrera et al.,2010, Responses of shelter and pet dogs to an unknown human) and social behavior in both species is flexible and dependent on experiences during the sensitive period. The only constraints I could see are those around length of sensitive period (which is longer in the domestic dog than domestic cat) but it is unclear to me how brain/body mass relate. Please elaborate on this a little more.

This is an extremely good point, the discussion of which I think could form the basis of a whole review paper in itself. For simplicity and to avoid adding further length to what is already a large review, I’ve removed this sentence

  1. L619-621: With the statement, “...similar evidence of genotypic-phenotypic relationships and their positive human-selection is currently lacking in domestic cats” I would prefer the wording be changed. To my knowledge, no work has been conducted in this area. So it is less about lacking evidence and more about lacking research.

I’ve added the following caveat to this sentence “While in domestic cats, this area of research has received less attention…” however I would argue that the studies discussed in the following section do (to a limited extent) address this topic and provide (some) evidence to the contrary of that reported in dogs, therefore I would prefer to leave the rest of the current wording in place. With edits, this section now reads:

Ln 706: “While in domestic cats, this area of research has received less attention, similar evidence of genotypic-phenotypic relationships for prosocial behavior and their positive human-selection is currently lacking.

For example, in (32) genetic surveys on microsatellite polymorphisms in a popula-tion of domestic cats were conducted, focusing on those linked to receptors (associated with oxytocin release) and previously linked with sociability in other species ( 128, 131). While (32) found positive associations between microsatellite length and the caretaker-rated friendliness of cats, pedigree cats were found to have shorter alleles compared to non-pedigree cats, implying a less enhanced genetic predisposition for human-sociability in pedigree populations. Additionally, while levels of oxytocin and cortisol have previously been found to increase in domestic dogs following social interactions with their owners (132), a recent preliminary study in domestic cats reported that both oxytocin and cortisol levels were actually decreased following human-social interactions (134).”